# Dual Disseminated Aspergillosis and Mucormycosis Diagnosed at Autopsy: A Report of Two Cases of Coinfection and a Review of the Literature

**DOI:** 10.3390/jof9030357

**Published:** 2023-03-15

**Authors:** Jason Murray, Zhen A. Lu, Karin Miller, Alex Meadows, Marissa Totten, Sean X. Zhang

**Affiliations:** 1Department of Pathology, Johns Hopkins University School of Medicine, Baltimore, MD 21287, USA; 2Department of Pathology, Children’s Hospital Los Angeles, Los Angeles, CA 90027, USA

**Keywords:** aspergillosis, mucormycosis, coinfection, autopsy, histology

## Abstract

Coinfection with invasive aspergillosis and mucormycosis in immunocompromised patients has been reported but is rarely confirmed by tissue histology or autopsy. Serum fungal biomarkers and culture are the primary diagnostic tools but are suboptimal for detecting fungal coinfection. Here, we present the cases of two patients who were immunocompromised due to hematologic malignancy where disseminated aspergillosis and mucormycosis coinfection was only diagnosed upon autopsy despite extensive fungal diagnostic workup, and also review recent literature of such instances of coinfection.

## 1. Introduction

*Aspergillus* species are common fungi found in natural and human environments and are usually benign colonizers of immunocompetent individuals, but can cause invasive aspergillosis in immunocompromised patients. In infected tissues, the genus of the fungus is identified histologically by filamentous septated hyphae with narrow angle branching. The order of Mucorales are commonly found in dirt and decaying plant matter and they contain several medially important fungi (e.g., *Rhizopus* sp., *Mucor* sp., *Rhizomucor* sp., *Lichtheimia* sp., *Cunninghamella* sp., etc.) that can cause angioinvasive mucormycosis. They are identified histologically in infected tissues by broad, ribbon-like, non-septated hyphae with wide angle branching.

While these otherwise benign fungi, ubiquitously found in human environments, can cause invasive fungal infections in immunocompromised patients [1], the clinical tools for fungal diagnostics are unfortunately limited. Diagnosis of fungal infection requires an astute workup and a broad differential. Serum fungal biomarkers such as galactomannan (GM) and 1,3-beta-D-glucan (BDG) are useful for identifying aspergillosis and some other mold infections, but not mucormycosis [2,3]. Furthermore, these markers are often insensitive. Fungal culture requires lengthy incubation to carry out and is often complicated by the presence of benign colonizing symbiotes. Similar issues complicate the use of molecular methods (PCR, DNA sequencing) as diagnostic aids. This makes fungal coinfection difficult to assess without histologic evidence of invasion and tissue destruction.

Given the diagnostic difficulty, it is not surprising that patients affected by disseminated fungal coinfection face extremely poor clinical outcomes with a high degree of morbidity and mortality. Here, we present the clinical courses and autopsy findings of two patients suffering from hematologic malignancy where first diagnosis of disseminated coinfection was made by autopsy and species determination was made by DNA sequencing from formalin-fixed paraffin-embedded (FFPE) tissues. 

We also performed a literature review by searching PubMed to identify instances of coinfection with both *Aspergillus* and Mucorales. Over 70 reported cases of *Aspergillus* and Mucorales coinfection were identified in this manner. However, only six cases demonstrated autopsy-proven aspergillosis and mucormycosis coinfection [4,5,6,7,8,9]. We summarized the reviewed cases where an autopsy was performed and which provided histologic evidence of disseminated dual fungal infection, as well as the cases we present, in Table 1 below.

## 2. Materials and Methods

Informed consent for the autopsies was given by the deceased patient’s next of kin. The autopsies were performed with standard bitemporal and Y-incisions. En-bloc evisceration was performed by AM, with subsequent dissection and tissue collection performed by JM, ZL, and KM. The organs were then separated anatomically into neck, chest, abdominal, and genitourinary blocks. Individual organs were then dissected. The lungs were inflated via the mainstem bronchus with 10% neutral buffered formalin to allow fixation prior to sectioning. Luminal organs were opened, and the mucosal and serosal surfaces evaluated. Solid organs and lungs were bread-loafed to evaluate for possible lesions. The brain was fixed in 10% neutral buffered formalin for 2 weeks before being sectioned under the guidance of a neuropathologist. Tissue sections were obtained from each organ, focusing on grossly identified lesions if present, and representative sections of grossly normal tissue if not. The tissue sections were fixed in 10% neutral buffered formalin for an additional 24–48 h and submitted to the Johns Hopkins Hospital Histology Laboratory for routine processing, embedding, and staining with hematoxylin and eosin (HE). Additional stains, periodic acid–Schiff stain (PAS), and Grocott methenamine silver (GMS), were also performed by the Histology Laboratory using standard methods. 

For Case 1, fungal DNA sequencing identification from formalin-fixed paraffin-embedded (FFPE) tissues was performed using targeted next-generation sequencing (amplifying ITS [internal transcribed spacer] region and D2 region of 28S) on Vela Diagnostics’ Sentosa Platform (SX101, SQ301, SQ Reporter). For Case 2, DNA was extracted from FFPE blocks using Zymo Fungal DNA extraction kits, followed by PCR reaction targeting ITS and D1D2 regions and Sanger sequencing (Applied Biosystems, ThermoFisher, Waltham, MA, USA). The sequence results were analyzed and reported by SmartGene IDNSTM Integrated Database Network System [10]. 

The cases were reviewed for the preparation of this manuscript and submitted for publication in accordance with Johns Hopkins Institution Review Board (IRB NA_00071002).

## 3. Cases

### 3.1. Case 1

Patient 1 was generally healthy until diagnosed with acute B-cell lymphoblastic leukemia (B-ALL) at age 40. Over the next 6 months, she underwent chemotherapy followed by a non-myeloablative preparative regimen of total body irradiation (TBI) before receiving an allogenic peripheral blood stem cell transplant (PBSCT). Unfortunately, she relapsed 7 months later. She achieved remission following chemotherapy and radiation treatment and was started on maintenance therapy before receiving TBI and a second PBSCT, 7 months after relapse.

In the period leading up to her second transplant, and for the initial 3 weeks post-transplant, the patient was neutropenic with an absolute neutrophil count (ANC) < 500/mm^3^ and occasionally agranulocytic (ANC < 100/mm^3^) after which she experienced neutrophil count recovery into a normal range (1500/mm^3^) during her final 3 weeks of life. Immediately after the second PBSCT, she began experiencing neutropenic fever despite prophylactic treatment with cefdinir, fluconazole, valacyclovir, and sulfamethoxazole-trimethoprim. Her antimicrobial treatment was initially widened to include cefepime followed by vancomycin after testing positive for *C. difficile*. A few days later, she began experiencing widespread signs and symptoms of fluid overload, thought to be due to sirolimus toxicity which was stopped and switched to tacrolimus.

Despite a month of steady improvement in controlling her fluid balance, Patient 1 then developed intermittent delirium and worsening mental status. An MRI of the head noted a 1.5 cm lesion in the left parietal cortex with a broad differential. Lumbar punctures showed no evidence of infectious processes or disease relapse histologically or with extensive laboratory testing (clear and colorless, 0 red blood cells, 0 white blood cells, glucose 67 mg/dL, protein 16 mg/dL, negative PCR for HHV-6, JCV, VZV, HSV, EBV, CMV, negative fungal culture, and negative culture for TB). At this point, her family decided to transition to comfort care measures and she passed away the following day. Of note, GM and BDG assays performed on the patient’s serum were negative prior to both PBSCTs and 8 days prior to her death. However, repeat testing was performed the day prior to death and returned positive results (Platelia GM index of 1.47, Fungitell BDG of 148 pg/mL) approximately 6 h after the patient passed.

At autopsy sectioning, the lungs revealed heterogeneous cut surfaces with innumerable firm, pale-tan, irregularly bordered lesions in the parenchyma. Also noted were distinct and darker reddish lesions involving and filling the vasculature. These lesions were histologically found to be fungi with morphological features consistent with both Aspergillus-like septate hyphae and mucoralean-like aseptate hyphae, respectively (Figure 1A–C).

Further dissection revealed gross abnormalities in the stomach, pancreas, left kidney and adrenal, bladder, brain parenchyma, and leptomeningeal vessels (Figure 1D). These grossly identified lesions were histologically consistent with invasive mucormycosis. Although the remaining organs and tissues were grossly unremarkable, histologic examination revealed invasive mucormycosis involving the serosal surfaces and serosal blood vessels of the small and large intestines, the uterine serosa and myometrium, and a randomly submitted lymph node (Figure 1E). The esophagus, liver, spleen, and ovaries showed no evidence of fungi grossly or histologically.

Fungal species identification performed on the FFPE tissues using targeted (ITS and D2) next-generation sequencing revealed that the fungus present with *Aspergillus*-like septate hyphae in the lung was *Aspergillus fumigatus*, whereas the fungus present with mucoralean-like aseptate hyphae in the lung, brain, small bowel, pancreas, and stomach was *Cunninghamella bertholletiae*. 

### 3.2. Case 2

Patient 2 had advanced Parkinson’s disease (PD) when she was diagnosed with FLT3-ITD-related acute myeloid leukemia (AML) at age 62. She was treated with a single round of chemotherapy, achieving remission. However, she was unable to tolerate further treatment due to her PD and relapsed 8 months later. Following 2 months of low-dose chemotherapy, she developed neutropenic fever and was found to have positive blood cultures with *Fusobacterium*, subsequently treated with piperacillin/tazobactam. She was also receiving prophylactic valacyclovir and fluconazole. During this time, she twice had negative GM and BDG serum assays. She then began receiving compassionate use of an investigational FLT3 inhibitor and remained chronically agranulocytic (ANC ranging from 0–420/mm^3^) in the final 4 months of her life.

A month later, Patient 2 developed vancomycin-resistant *Enterococcus faecium* sepsis and received linezolid. CT scans at this time showed no mass in the patient’s chest, abdomen, or pelvis. Over the following 2 months, she had several short-interval ED presentations and hospital admissions for vomiting, poor oral intake, hyponatremia, possible syncope, and was treated with IV hydration and cessation of the FLT3 inhibitor. On her final admission, the day before she passed, a CT showed a mass-like density in the left lower lung, nonspecific density in the right lung fissure, low-density hepatic and splenic lesions, and a distended stomach and multiple distended proximal to mid small bowel loops, compatible with a small bowel obstruction. These findings were felt to be consistent with soft-tissue spread of her hematologic disease.

In the setting of small bowel obstruction, nasogastric (NG) intubation was attempted. Unfortunately, during NG tube placement she experienced a suspected aspiration event. Due to poor oxygenation and a new vasopressor requirement, she was intubated and transferred to the ICU where she also received empiric vancomycin and piperacillin/tazobactam. The following day, she suddenly became extremely hypoxic on the ventilator, developed asystolic cardiac arrest, and passed away despite resuscitation efforts.

The autopsy showed a 3 cm white, firm area in the lower lobe of the left lung. Histologically, this area showed angioinvasive fungus morphologically compatible with *Aspergillus*-like species (Figure 1F).

In the small bowel, a 5 cm area of thickening was noted grossly, consistent with the small bowel obstruction noted radiologically. In the thickened area, angioinvasive fungus morphologically consistent with Mucorales fungi was identified (Figure 1G). Angioinvasive fungus with this same morphology was also found in the liver (Figure 1H). 

Additionally, invasive fungal species were also found in the spleen. However, morphologic evaluation was limited by the extent of necrosis.

Subsequent pan-fungal PCR/sequencing (ITS and D1D2) identified the fungus present with *Aspergillus*-like hyphae in the lung as *Aspergillus fumigatus* and the fungus present with mucoralean-like aseptate hyphae in the small intestinal tissue as *Mucor indicus*.

## 4. Discussion

Disseminated fungal infections are often fatal and diagnosis is complicated by the severity of illness. Patients with such infections are often complex and can be clinically unstable, poorly able to tolerate invasive procedures. Clinical teams often rely on GM or BDG assays or culture of blood, BAL, or CSF samples to assess for possible fungal infection in these situations, despite the insensitivity of such assays, rather than an invasive tissue biopsy in clinically unstable patients [2]. Despite the widespread nature of disseminated fungal infection in our patients, GM and BDG assays yielded repeated negative results. In the case of Patient 1′s disease, both GM and BDG were negative 2 months prior to her death and negative again 8 days prior to her death. Repeat assays collected 1 day before her death finally yielded positive results (Platelia GM 1.47; Fungitell BDG 148 pg/mL) several hours after she died. Such delays in diagnosis are not uncommon with the limitations of fungal assays, and can be detrimental to patients.

Radiographic imaging can be similarly insensitive, as demonstrated again in the case of Patient 1, where no targetable lesion was present on imaging to enable a tissue biopsy until an MRI of the head demonstrated a brain lesion less than 2 days before her demise. In light of the poor clinical course, new brain lesion, and negative CSF results, the patient had been transitioned to comfort care measures only by her family. Daily chest X-rays in the period prior to her death showed no abnormality aside from pleural effusions, and thoracentesis was negative for fungus on cytologic histology. An ultrasound of the liver performed less than 10 days before passing showed a normal pancreas, which was ultimately shown to be heavily involved. This issue was also seen with Patient 2, where a CT chest/abdomen/pelvis performed 2 months before she passed showed no lesions, and a CT performed the day before she passed identified lesions in the small intestine, lungs, liver, and spleen. Unfortunately, the patient passed the following day, before a biopsy could be performed.

Such situations are not unique and in many of the reviewed reports, including Patients 3, 5, 6, and 7, serum and BAL assays of GM, serum assay of BDG, as well as fungal cultures taken from BAL or sometimes even biopsy tissue, were initially negative and only became positive later in the treatment course (Table 1). Even when such assays yield positive results, tissue histology remains the gold standard for diagnosis as the visual confirmation of hyphae invading into tissue rules out contaminants and benign colonizing microbes. Histology offers further benefits in cases of co-infection, such as those described here, as Mucorales fungi are not identifiable by current serum markers and diagnosis must be made by histology or molecular methods [3]. Thus, even after invasive fungal infection is suspected or proven, coinfection should be considered, especially when clinical improvement does not match the diagnosis/treatment, since it is not uncommon and may be under-recognized.

While our patients and several of those presented in Table 1 were immunosuppressed as treatment for hematologic malignancy or organ transplantation, Patients 5 and 7 suffered from autoimmune diseases. In reviewing the literature, many patients with diabetes mellitus, chronic steroid use for rheumatologic diseases, and short-term use of high-dose steroids to treat SARS-CoV-2 in otherwise immunocompetent patients were found to experience fungal coinfection [11,12,13]. While these cases were not included due to lack of autopsy results, they indicate that fungal coinfection is not uncommon and impacts a broad patient population.

Autopsies offer a unique ability to quantify the extent of fungal dissemination by sampling tissues that were not previously biopsied, and characterizing lesions not identified on imaging [14,15]. The identification of previously unknown disseminated fungal co-infection in our patients, and Patients 4 and 8 in the reviewed cases, highlights the value of autopsy in providing diagnostic information, and indicates the role autopsy could have in epidemiologic studies to characterize the true extent and frequency of these infections. Furthermore, the tissue available at autopsy allows for extensive characterization and workup that is often impossible with tissue obtained from a small biopsy. In our cases, DNA sequencing supported the histologic finding of multiple fungal morphologies and allowed for precise speciation of the fungal pathogens in tissues without the need to perform cultures.

## Figures and Tables

**Figure 1 jof-09-00357-f001:**
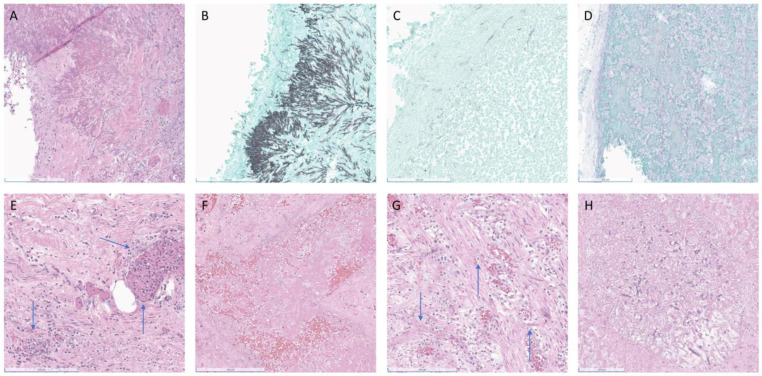
(**A**–**E**) (Patient 1), (**F**–**H**) (Patient 2). (**A**) Patient 1, HE left upper lobe (LUL) with narrow septate hyphae and narrow angle branching consistent with Aspergillus, (**B**) GMS of LUL consistent with Aspergillus, (**C**) GMS of LUL with area of broad aseptate ribbons and wide branching consistent with Mucorales, (**D**) PAS left parietal leptomeninges, (**E**) HE uterus, (**F**) Patient 2, HE left lower lobe of lung, (**G**) HE small bowel, (**H**) HE liver. Blue arrows localize the more difficult to identify fungal hyphae. Scale bar: 200 μm.

**Table 1 jof-09-00357-t001:** A literature review was performed by searching PubMed for combinations of *Aspergillus*/aspergillosis, *Mucor*/Mucorales/mucormycosis, and mixed mycosis, mixed mycotic infection, coinfection or dual infection to identify relevant papers in English. Cases referenced in the papers initially retrieved were also collected. Over 70 reported cases of *Aspergillus* and Mucorales coinfection were identified in this manner. Cases were excluded from the table based on (1) if the patient survived, (2) if an autopsy was not performed after patient passed, (3) if infection was of the superficial skin or due to trauma, and (4) if the paper did not provide images of the fungal histology. Patient 1 and 2 are the cases described in our paper. Fungal cultures were taken at time of autopsy unless otherwise stated. Fungal culture and biomarker timing are when the specimens were obtained, not when the results were available to clinicians. ND = not done, GM = galactomannan, BDG = beta-D-glucan, BAL = bronchoalveolar lavage, CSF = cerebrospinal fluid, AmB = amphotericin B, L-AmB = liposomal amphotericin B.

Patient	Age	Gender	Underlying Condition	Immunosuppressive Treatment	Antifungal Treatment	Autopsy Evidence of Dual Invasive Fungal Infections		Fungal Identification (Genus/Species)		Fungal Biomarkers	
						Aspergillosis	Mucormycosis	By Tissue Culture	By DNA Sequencing on FFPE Block	GM	BDG
1	41	F	B-ALL, sirolimus related pericardial effusion	chemotherapy, radiation, and steroids	none	lungs	lungs, brain, stomach, small bowel, large bowel, kidney, bladder, uterus, pancreas	ND	*A. fumigatus* (Lung), *C. bertholleitiae* (lung, brain, small bowel, pancreas, stomach)	neg (8 d prior to death); 1.47 (serum; 1 d prior to death)	neg (<31 pg/mL, 8 d prior to death); 148 pg/mL (1 d prior to death)
2	63	F	AML, small bowel obstruction	clinical trial of FLT3 inhibitor	none	lungs	lungs, small bowel, liver, spleen	ND	*A. fumigatus* (lung), *M. indicus* (small bowel)	ND	ND
3	79	M^4^	CLL	chlorambucil and obinutuzumab; corticosteroids	none	lungs	heart, lymph node, thyroid, kidney, brain	*A. fumigatus* (lungs) from autopsy tissue	*L. corymbifera* (heart)	neg (~8 weeks prior to death), >9.3 immediately prior to death	neg (~8 weeks prior to death), neg (immediately prior to death)
4	19	M^5^	ALL	presumed chemotherapy for ALL	oral AmB	lung, brain	lung, liver, spleen	*A. fumigatus* (lung, brain), *Rhizopus* spp. (lung, liver, spleen) from autopsy tissue	ND	ND	ND
5	55	F^6^	dermatomyositis	prednisone	voriconazole, L-AmB	brain	brain	ND	ND	pos (blood, 11-months before death), pos (BAL, 10-month before death), pos (CSF, 5 month before death)	ND
6	52	M^7^	heart transplant	transplant immunosuppression	none	lung	lung	*A. fumigatus* (lung biopsy 60 d post-transplant); *A. fumigatus* and *Mucor* spp. (15 w post-transplant)	ND	3.3 (60 days post-transplant)	ND
7	70’s	F^8^	p-ANCA vasculitis	steroids and azathioprine	fluconazole, micafungin, L-AmB	lungs	lungs, heart	ND	ND	ND	pos (7 days prior to death)
8	61	M^9^	hematologic malignancy	prednisolone	unknown	lung	lung	ND	ND	ND	ND

## Data Availability

Sequencing data and scanned histology slides, and tissue blocks can be provided upon request.

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
