# Peer review of "Dual Disseminated Aspergillosis and Mucormycosis Diagnosed at Autopsy: A Report of Two Cases of Coinfection and a Review of the Literature"

_jof, 2023, doi:10.3390/jof9030357_

Round 1
Reviewer 1 Report
The clinical description lacks relevant information:
Case 1: The timing and duration of agranulocytosis for
Case 2: Clarification of the notion of neutropenia (neutropenia or agranulocytosis??)
Did the patients receive antifungal prophylaxis? If yes, which one.
From which environment were issued the patients (city, country, profession.. exposition to fungi...)
Regarding the Pubmed Review: Precise the keywords that were used to identifiy cases. Was Search limited to englisch language? Was is limited to immunosuppressed patients?
Reviewer 2 Report
This is a well written article highlighting co-infection with Aspergillus and Mucormycosis. The literature review also adds to the value of the manuscript.
The authors may highlight, why despite being a gold standard, tissue biopsies in both of the patients were not performed while patients were alive.
Reference 10 and 15 are incomplete.
